# Neoadjuvant Treatment Lowers the Risk of Mesopancreatic Fat Infiltration and Local Recurrence in Patients with Pancreatic Cancer

**DOI:** 10.3390/cancers14010068

**Published:** 2021-12-23

**Authors:** Sami-Alexander Safi, Lena Haeberle, Alexander Rehders, Stephen Fung, Sascha Vaghiri, Christoph Roderburg, Tom Luedde, Farid Ziayee, Irene Esposito, Georg Fluegen, Wolfram Trudo Knoefel

**Affiliations:** 1Department of General, Visceral, Thoracic and Pediatric Surgery (A), Medical Faculty, Heinrich-Heine-University, University Hospital Duesseldorf, Moorenstr. 5, 40225 Duesseldorf, Germany; sami-alexander.safi@med.uni-duesseldorf.de (S.-A.S.); Rehders@med.uni-duesseldorf.de (A.R.); Stephen.Fung@med.uni-duesseldorf.de (S.F.); Sascha.Vaghiri@med.uni-duesseldorf.de (S.V.); Georg.Fluegen@med.uni-duesseldorf.de (G.F.); 2Institute of Pathology, Medical Faculty, Heinrich-Heine-University, University Hospital Duesseldorf, Moorenstr. 5, 40225 Duesseldorf, Germany; LenaJulia.Haeberle@med.uni-duesseldorf.de (L.H.); Irene.Esposito@med.uni-duesseldorf.de (I.E.); 3Department of Gastroenterology, Hepatology and Infectious Diseases, Medical Faculty, Heinrich-Heine-University, University Hospital Duesseldorf, Moorenstr. 5, 40225 Duesseldorf, Germany; Christoph.Roderburg@med.uni-duesseldorf.de (C.R.); Tom.Luedde@med.uni-duesseldorf.de (T.L.); 4Department of Diagnostic and Interventional Radiology, Medical Faculty, Heinrich-Heine-University, University Hospital Duesseldorf, Moorenstr. 5, 40225 Duesseldorf, Germany; Farid.Ziayee@med.uni-duesseldorf.de

**Keywords:** mesopancreas, PDAC, CRM, mesopancreatic excision, peripancreatic tissue, neoadjuvant

## Abstract

**Simple Summary:**

After the implementation of an in-depth histopathological pancreas protocol, curative resection rates for pancreatic head cancers have drastically dropped. Standardized extended resections using embryo-anatomic landmarks (MPE), have recently been prooved to increase margin-negative resection rates. The mesopancreatic fat, excised during these extended resections, was infiltrated in the majority of the patients. Neoadjuvant treatment is an emerging topic of interest for pancreatic cancer patients. It remains unclear if these extended resections are still warranted in patients after neoadjuvant treatment. Neoadjuvant treatment lowered the risk for mesopancreatic fat infiltration and patients were less prone to local recurrence and margin positive resections when compared to patients after upfront surgery. However, the majority of the patients are yet diagnosed with mesopancreatic fat infiltration, justifying this extended approach synergistically with the treatment strategies for colorectal cancer.

**Abstract:**

Background: Survival following surgical treatment of ductal adenocarcinoma of the pancreas (PDAC) remains poor. The recent implementation of the circumferential resection margin (CRM) into standard histopathological evaluation lead to a significant reduction in R0 rates. Mesopancreatic fat infiltration is present in ~80% of PDAC patients at the time of primary surgery and recently, mesopancreatic excision (MPE) was correlated to complete resection. To attain an even higher rate of R0(CRM−) resections in the future, neoadjuvant therapy in patients with a progressive disease seems a promising tool. We analyzed radiographic and histopathological treatment response and mesopancreatic tumor infiltration in patients who received neoadjuvant therapy prior to MPE. The aim of our study was to evaluate the need for MPE following neoadjuvant therapy and if multi-detector computed tomographically (MDCT) evaluated treatment response correlates with mesopancreatic (MP) infiltration. Method: Radiographic, clinicopathological and survival parameters of 27 consecutive patients who underwent neoadjuvant therapy prior to MPE were evaluated. The mesopancreatic fat tissue was histopathologically analyzed and the 1 mm-rule (CRM) was applied. Results: In the study collective, both the rate of R0 resection R0(CRM−) and the rate of mesopancreatic fat infiltration was 62.9%. Patients with MP infiltration showed a lower tumor response. Surgical resection status was dependent on MP infiltration and tumor response status. Patients with MDCT-predicted tumor response were less prone to MP infiltration. When compared to patients after upfront surgery, MP infiltration and local recurrence rate was significantly lower after neoadjuvant treatment. Conclusion: MPE remains warranted after neoadjuvant therapy. Mesopancreatic fat invasion was still evident in the majority of our patients following neoadjuvant treatment. MDCT-predicted tumor response did not exclude mesopancreatic fat infiltration.

## 1. Introduction

PDAC is estimated to become the second leading cause of cancer-related deaths by 2030 [1,2]. Kausch et al. and Whipple et al. first described a regional resection of the pancreatic head in the early 20th century [3,4]. These procedures have become the gold standard in surgical treatment for ductal adenocarcinomas of the pancreas (PDAC) and the defining anatomic landmarks during pancreatic resection have remained largely the same during the past century.

Even though most patients present with locally or systemically advanced disease and are thus not candidates for primary resection, chemotherapeutic regimens, including gemcitabine and its combinations, have not resulted in improved 5-year overall survival (OS) [5,6,7]. Recently, the implementation of modified FOLFIRINOX (5-fluorouracil, leucovorine, irinotecan, and oxaliplatin) as an extended adjuvant therapy has led to some improvements [8], but the prospects for inoperable patients remain dismal.

The majority of PDAC patients traditionally suffered local recurrence during follow-up, contrasting with the high reported margin-negative resection rates [9,10,11,12]. This observation led to evolved histopathological standards, culminating in the implementation of the circumferential resection margin (CRM) in 2004 [13,14]. This modified pathological protocol includes the evaluation of the ventral and dorsal pancreatic surfaces, as well as the medial pancreatic margin (i.e., the groove of the superior mesenteric vein and the surface facing the superior mesenteric artery) [15,16]. Widespread implementation of the CRM resulted in plummeting margin-negative resection rates, explaining the high incidence of local recurrences, and thus indicating that a more radical surgical approach may lead to improved local tumor control and better long-term results in pancreatic cancer patients [14,17].

However, the current resectability criteria are solely defined by the vascular infiltration, representing the medial resection margin. Yet, the dorsal resection site constitutes the second highest rate of tumor infiltration, indicating that for local tumor control, a tumor-free circumferential resection of the complete pancreatic head is paramount.

Total mesorectal excision, a resection technique that utilizes a predefined anatomic space, has provided a significant improvement in local tumor control and survival outcome for rectal cancer patients [18,19]. This technique of using predefined anatomic landmarks for complete resection has been translated to mesocolic excision for patients with colon cancer and has resulted in a similar oncological benefit [19].

Just in the past few years, spearheaded by the Japanese pancreatic society, the concept of mesopancreatic excision (MPE) has been developed [20]. Our group was able to demonstrate that in ~80% of resectable PDAC patients, the mesopancreatic fat was infiltrated [17]. The invasion front remains the most potent risk factor for local recurrence, as vital tumor cells are more frequently observed here, compared to the tumor center [21,22]. Not only was the infiltration of the mesopancreatic fat evident at the medial resection margin, but also at the dorsal resection margin, underlining the pathological results after CRM implementation. To achieve maximum rates of margin clearance, MPE during structured PDAC resection is thereby justified. Nevertheless, the rates of true margin-negative resections (R0(CRM−)), while improved, still remain low, reaching just ~50% in PDAC patients.

Two factors could contribute to the yet inadequate degree of margin clearance: (1) current resectability criteria only partially reflect the local tumor burden, as the dorsal resection site is not considered in the redefined ABC-criteria; (2) a lack of downsizing therapeutic options, as employed in the treatment of locally advanced rectal carcinomas.

Our group was previously able to demonstrate that mesopancreatic fat stranding, evaluated by preoperative diagnostic imaging, correlated with the actual mesopancreatic fat infiltration, independent on the vascular status of the patient [23]. These novel MDCT parameters could be utilized for more precise resectability criteria.

To date, neoadjuvant therapy is only recommended for synchronously metastasized patients or borderline resectable cases [24,25,26,27,28,29]. Yet, patients who are deemed resectable in preoperative diagnostic imaging still have a high chance of insufficient tumor clearance (R1/R0(CRM+)), allowing the argument that neoadjuvant therapy may be able to reduce tumor burden and increase the number of complete resections even in primarily resectable patients [30,31]. It may thus be only a matter of time until neoadjuvant treatment will be the standard of care in the majority of PDAC patients.

It remains unclear if vital tumor cells routinely remain in the mesopancreatic lamina even after MPE following neoadjuvant therapy. The aim of this study was to histopathologically evaluate the mesopancreatic fat in patients who received MPE during PDAC resection following neoadjuvant therapy, in order to quantify if this thorough surgical approach is still warranted. Preoperative multi-detector computed tomography (MDCT) slides were reevaluated for treatment response after neoadjuvant treatment and histopathological specimens were revisited.

## 2. Materials and Methods

### 2.1. Patient Selection and Demographic Data

All patients who underwent pancreatic surgery including MPE for any PDAC with curative intent, irrespective of tumor location, stage and microscopic resection margin, at the University Hospital of Duesseldorf between 2010 and 2021 were screened for inclusion in this study from a prospectively maintained database. Inclusion criteria were surgically resected ductal adenocarcinomas of the pancreas (PDAC) with neoadjuvant therapy and sufficient information on follow-up examinations. Patients who underwent surgery for periampullary lesions other than PDAC were excluded from the study. Patients who received upfront surgery for PDAC without neoadjuvant therapy (including MPE) during the same study period served as a control group for correlation analysis of mesopancreatic fat infiltration and resection margin status [17]. TNM staging, grading, perineural invasion as well as lymphatic and venous invasion were obtained from the original pathological reports. Histopathological slides were re-visited by an experienced pathologist for pancreatic cancer, with focus on treatment response and mesopancreatic fat invasion, in order to re-evaluate the resection margins. Staging system was updated to the 8^th^ Edition of the UICC TNM classification of malignant tumors [32]. Clinico-pathological data were reviewed. The study was carried out in accordance to the guidelines of Good Clinical Practice and the Declaration of Helsinki. The study was approved by the Institutional Review Board (IRB) of the Medical Faculty, Heinrich Heine University Duesseldorf (IRB-no. 2019-473_2).

### 2.2. Radiographic Imaging

Preoperative multiphasic multi-detector CTs (MDCT) following neoadjuvant therapy were available for re-evaluation. The CTs were retrospectively analyzed by experienced hepatopancreaticobiliary radiologists, blinded for resection status and postoperative staging. Local tumor response was analyzed [23]. (Figure 1). Scoring criteria of tumor response was scored by the mesopancreatic attenuation of the fatty tissue in terms of diminished/decreased mesopancreatic fat stranding and/or decreased tumor size with an smaller encasement of the SMA or celiac trunk/common hepatic artery. Due to the limited number of enrolled patients, no subgroup analyses (MPS1-3) was performed. Patients were subgrouped into tumor response or no tumor response.

### 2.3. Operative Procedure

The operative procedure performed for PDAC, which utilizes embryo-anatomic landmarks, was already described by Safi et al. [17]. In summary, a wide Kocher maneuver is performed in order to achieve a simultaneous transection of the mesopancreatic lamina followed by a complete para-aortic and interaortocaval lymphadenectomy to the right border of the superior mesenteric artery (SMA) and celiac trunk (CT) (Figure 1). When entering the abdomen by a transversal upper laparotomy, the right colon flexure and ascending colon are mobilized by transecting the right Toldt’s fascia, which is extended up to Gerota’s fascia and continued medially underneath the mesopancreas with the dissection of the fusion fascia of Treitz. This is continued up to the origin of the Treitz ligament. The dissection is then accomplished to the inferior border of the pancreatic neck, visualizing the portal vein/superior mesenteric vein (PV/SMV). Following this, dissection of the hepatoduodenal ligament (left and right hepatic artery, common hepatic artery (CHA), gastroduodenal artery (GDA), common bile duct, and portal/superior mesenteric vein (PV/SMV)) completes surgical exploration. Lymphadenectomy and dissection of the common hepatic artery is performed up to its origin from the celiac trunk (CT). If resectability criteria were met, the jejunum, the ligament of Treitz and the duodenal bulb (or distal stomach) can then be transected (“Point of no Return”). The jejunum is then mobilized to the patient’s right side. After the pancreatic head is completely separated from the PV/SMV and the SMA, the pancreatic neck is divided. Next, complete lymphadenectomy and dissection of the PV/SMV is completed. If a possible tumor infiltration is present, venous resection and reconstruction is routinely performed. Sharp preparation along the SMA and the CT up to their aortic origins is carried out, dissecting perivascular lymphatic tissue. To avoid persistent diarrhea only 180° to 270° of the right circumference of the SMA are dissected. If cancerous involvement is intraoperatively suspected, dissection of the SMA is extended to the left circumference (Figure 2).

All resections were performed by trained hepatobiliary surgeons of our department. In summary, the aim of the procedure is a complete dissection of perineural and lymphatic tissue including the fat surrounding the pancreatic head/uncinate process (CHA, GDA, CT, SMA, PV, SMV) in an “en-bloc” resection (Figure 3A,B).

### 2.4. Pathological Analysis

The CRM evaluation was implemented at the University Hospital of Duesseldorf in September 2015. The oral/aboral duodenal, bile duct and pancreatic neck resection margin, as well as the dorsal resection margin and, if applicable, portal vein specimen were examined according to the LEEPPs pathological protocol. Additionally, the mesopancreatic adipose tissue was histopathologically evaluated for cancerous infiltration (Figure 4A,B). Cases evaluated before 2015 were re-visited by an experienced pancreaticobiliary pathologist and, if sufficient slides were available, a CRM status including evaluation of the mesopancreatic fat was designated. This included the evaluation not only of the dorsal, but also ventral and medial CRM. In addition, the “1-mm rule” was implemented: a minimum margin clearance of 1 mm defined R0CRM−, whereas margin clearances between 0 and 1 mm were judged as R0(CRM+) [33]. Tumor response following neoadjuvant treatment was graded according to the College of American Pathologists (CAP) (grade 0 = complete response, grade 1 = near complete response, grade 2 = partial response and grade 3 = poor/no response) (Figure 5A–D) [34].

### 2.5. Postoperative Follow-Up

All patients were pre- and postoperatively evaluated and discussed in an interdisciplinary tumor board regarding adjuvant therapy and further procedure. If the follow-up examinations were performed at our institution, irrespective of the adjuvant treatment constellation, computed tomography of the thorax and abdomen was performed every 3 months for the first 2 years, followed by every 6 months thereafter. Patients with suspicious metachronous masses were again discussed in the tumor board for further therapy. In cases where follow-up procedures were performed at other institutions, the legal registration office was contacted for survival records of these patients.

### 2.6. Statistics

To compare the distribution and the influence of clinic-pathological variables, the chi-square test or fisher exact test was applied. Correlation analysis was performed using Pearson’s correlation coefficient method. Analyses were performed using SPSS statistics for Windows (version 26.0; SPSS, Inc., Chicago, IL, USA). Statistical significance was defined as *p* < 0.05.

## 3. Results

### 3.1. Demographic Data

A total of 27 patients met our inclusion criteria (11 females (40.7%) and 16 males (59.3%)). Table 1 summarizes Clinic-pathological characteristics of the cohort. All 27 patients received neoadjuvant treatment because of advanced disease. The median age of all patients at the time of surgery was 66 years (range 41–80 years). In total, three patients (11.1%) received gemcitabine mono therapy, while five patients (18.5%) received a combination therapy including gemcitabine and three patients (11.1%) received combined radiochemotherapy including gemcitabine. Fifteen patients (55.6%) were treated with modified FOLFIRINOX. The dosing regimen is stated in a Appendix A. One of 27 patients deceased during the first 30 postoperative days (Clavien-Dindo V; 30-day mortality rate: 3.7%). Median length of hospital stay (LOS) was 23 days (range: 12–153 days). All patients received MPE during pancreatic surgery. In all patients, a fibrous sheet was visible at the posterior resection site running between the duodenum and the origin of the SMA/CT (Figure 3A,B).

In all patients, preoperative MDCT scans following neoadjuvant therapy were performed to investigate local treatment response. In 17 (62.9%) patients, MDCT indicated a treatment response and these patients were re-staged from non-resectable to borderline resectable. In the other nine patients, MDCT after neoadjuvant treatment showed a stable disease. Out of the 27 patients, mesopancreatic fat stranding after neoadjuvant treatment was still visible in 20 patients.

### 3.2. Histopathological Results

Resection status:

Histopathological analyses and resection status are summarized in Table 1. In all 27 patients, detailed CRM was histopathologically evaluated. Fifteen patients (55.6%) were evaluated before 2015 and needed sophisticated histopathological re-evaluation, whereas in 12 patients (44.4%), CRM evaluation in the context of a standardized pancreatic protocol was primarily applied. When applying the 1 mm rule, true negative resection margins were still present after re-evaluation in 17 patients (62.9%). Of the remaining patients, four (14.8%) had insufficient tumor clearances (R1) and in six patients (22.2%), tumor residues were detected within the 1 mm margins (R0(CRM+)).

Neoadjuvant tumor response:

Histopathological re-evaluation of tumor response was performed on the complete study cohort (Figure 5A–D). In two patients (7.4%), complete tumor response (CAP grade 0) was detected. In five patients (18.5%), a near complete tumor response (CAP grade 1) was diagnosed, whereas in 15 (55.5%) and five (18.5%) patients, partial (CAP grade 2) and poor/no tumor response (CAP grade3) was evident.

Mesopancreatic evaluation:

In all 27 patients, paraffin-embedded histopathologic specimens were available for retrospective re-evaluation of the mesopancreatic fat tissue of the peripancreatic dorsal resection margin (Figure 4A,B). Tumor infiltration of adipose tissue was evident in 17 patients (62.9%), whereas in only 10 patients (37.1%), mesopancreatic adipose tissue had no tumor infiltration (Table 2). In all 17 patients with mesopancreatic fat infiltration, vital tumor cells were present.

### 3.3. Influence of Clinicopathological Variables

The results of the statistical analysis are summarized in Table 2, Table 3 and Table 4. Mesopancreatic fat infiltration was compared in patients with and without histopathological tumor response. Patients lacking mesopancreatic fat infiltration had a significantly better histopathological tumor response, compared to patients with mesopancreatic fat infiltration (* *p =* 0.003) (Table 2). This correlation was significant in the Pearson correlation (* *p* = 0.005, r = +0.585 **) (Appendix A). Resection status (R0(CRM−) vs. R1/R0(CRM+)) was compared with mesopancreatic histology. The rate of R0(CRM−) resection was 62.9% in the total cohort. For patients lacking mesopancreatic fat infiltration, this rate was significantly higher compared to patients with histopathological mesopancreatic fat infiltration (* *p* = 0.031; R0(CRM−) in MP− = 80.0% and R0(CRM−) in MP+ = 52.9%) (Table 2).

Patients with better treatment response had a significantly higher percentage of complete (R0(CRM−)) resections (** p =* 0.042). While patients with good histological treatment response (CAP grade 0 and 1) had a 100% rate of complete resections, only 57.1% and 20.0% of the patients with partial and poor/no response (CAP grade 2 and 3), respectively, received a R0(CRM−) resection. None of the R1/R0(CRM+) patients had a good histopathological response (CAP grade 0 and 1) (Table 3).

Mesopancreatic fat infiltration and tumor response was compared to preoperative MDCT variables (Table 4). Radiographically presumed tumor response did predict mesopancreatic fat infiltration (*p* = 0.042), whereas the trend for histopathologically verified treatment response was present but not statistically significant (*p =* 0.112 for treatment response) (Table 4).

### 3.4. Mesopancreatic Fat Infiltration in Patients with and without Neoadjuvant Therapy

During the study period (2010–2021), 173 patients received upfront surgery for primary resectable PDAC. In all patients, the mesopancreatic fat infiltration was analyzed. In 131 patients (75.7%), the mesopancreatic fat was histopathologically infiltrated. We detected a statistical different rate of mesopancreatic fat infiltration between primary resected patients and the 27 patients resected following neoadjuvant therapy (*p =* 0.039) (Table 5).

Follow-up data was available in the 27 neoadjuvant treated patients and the 173 patients who received upfront surgery (Table 6). Follow-up analysis revealed that systemic relapse was not prevented by degree of surgical radicality (*p =* 0.143). Irrespective of the treatment strategy, most patients were diagnosed with a systemic relapse during follow-up analysis (46.8% of the patients after upfront surgery and 69.2% of the patients after neoadjuvant treatment (*p* = 0.143). Neoadjuvant-treated patients showed a significantly lower rate of local recurrence during follow-up investigations when compared to patients after upfront surgery (*p =* 0.040) (7.4% vs. 16.8%). Thus, the negative effect of neoadjuvant treatment on the infiltration status of the mesopancreatic fat presumably resulted in a more secure local tumor control (Table 5 and Table 6).

## 4. Discussion

During the past few decades, the prognosis of PDAC patients did not improve significantly. One explanation may be suboptimal resection during surgery. Remarkably, refined histopathological assessment implementing the CRM-method, recently revealed tumor cells infiltrating the mesopancreatic fat in ~80% of patients, suggesting that a more thorough surgical approach might lead to an improved local tumor control [17]. In this context, mesopancreatic excision (MPE) was demonstrated to contribute to increased rates of R0(CRM−) resections [14,17,35,36], yet the overall rate remains poor compare to surgical results in other malignancies of the digestive system [37,38,39].

It is, therefore, conceivable that a neoadjuvant down-sizing concept, as established in esophageal and rectal cancer [40,41], could improve margin-negative resection rates in PDAC patients who are currently deemed primarily resectable [42,43]. Currently, it is mainly patients diagnosed with limited synchronous metastases or borderline resectable PDAC that are considered for neoadjuvant treatment [44]. There remains a significant lack of reliable data on the impact of neoadjuvant therapy in PDAC, as only limited and/or retrospective studies are available, making the results of two running prospective studies clearly warranted [31,45]. Furthermore, we showed that surgically resected patients with metastasized PDACs to the para-aortic lymph nodes showed a similar median overall survival both to patients with histopathologically verified negative para-aortic lymph node metastases and to patients who did not receive para-aortic lymph node dissection [46]. Thus, it could be appropriate to suggest that most patients who receive upfront surgery yet are diagnosed with an advanced stage of PDACs. Yet, primary margin clearance of PDAC remains the only surgical prerequisite for long term survival [17,47] and neoadjuvant therapy will thus likely play a major role in the near future [48].

However, it is unknown if MPE will further improve local tumor resection and if it is thus still necessary following neoadjuvant therapy. The aim of the current study was to quantify the rate of mesopancreatic fat infiltration applying the redefined histopathological standard in patients who received neoadjuvant therapy prior to pancreatic surgery and MPE [47].

Based on the results of this study, we reach the following conclusion: (1) mesopancreatic fat infiltration was still evident in the majority of the patients and it was a marker for CAP tumor response grading; (2) resection margin status mirrored the MP infiltration status and the CAP tumor response grading; (3) MDCT-predicted tumor response was significantly associated with MP infiltration status; (4) neoadjuvant treatment was significantly associated with risk of MP infiltration and local recurrence when compared to patients after upfront surgery.

In our cohort, mesopancreatic fat infiltration was still evident following neoadjuvant treatment in the majority of the patients. Most importantly, neoadjuvant treatment had a negative effect on mesopancreatic fat infiltration and patients were significantly less prone to extended tumor involvement after receiving neoadjuvant treatment. True margin-negative resections (R0CRM−) were achieved in over 62.9% of the patients, suggesting that a neoadjuvant concept in patients with an otherwise resectable mass could indeed contribute to a down-sizing and improved resection rates [31,45]. Although not reaching statistical significance due to the sample size, the rate of R0(CRM−) resections was higher in the neoadjuvant group (*n* = 27, R0(CRM−): 62.9%) when compared to patients who received upfront surgery during the same time at our institution (*n* = 173, R0(CRM−): 50.1%, *p* = 0.127). Vital tumor cells were detected in the resected retropancreatic fatty tissue in 62.9% of the neoadjuvant treated patients, indicating that neoadjuvant therapy alone may not be sufficient for local tumor control and MPE should still be performed.

Local recurrence during follow-up was detected in only 7.4% of the neoadjuvant treated patients, compared to 16.8% of the patients following upfront surgery. This is presumably due to the negative effect of neoadjuvant treatment on the mesopancreatic fat infiltration and the higher rate of R0CRM− resections.

One limitation of our study is the small cohort. Yet, when compared to other recent retrospective studies on neoadjuvant treatment [49,50], a similar number of patients were enrolled as in our analysis. Another limitation of our study is the retrospective and mono-institutional nature of the analysis. Patients received different neoadjuvant regimes in this study, further limiting our conclusions due to heterogeneity. However, as neoadjuvant treatment is still under investigation and current clinical trials have not demonstrated a significantly improved survival of neoadjuvantly treated borderline resectable patients when compared to patients after upfront surgery [31,45,51], the number of eligible patients is still limited. Furthermore, it remains unknown which multimodal regime is superior to the other in a neoadjuvant setting. Taking this small sample size into consideration, we did not perform a survival analysis, as its statistical relevance would be questionable. Nevertheless, a negative association was found between neoadjuvant treatment and local recurrence, concluding that multimodal therapy following MPE could contribute to local tumor control.

Our results emulate similar observations as in rectal cancer studies. The introduction of neoadjuvant radiochemotherapy and the standardization of total mesorectal excision [18], performed independent of the tumor response, resulted in significantly lower local recurrence rates, while overall survival remained similar to patients undergoing upfront surgery [40,52,53]. Recently published randomized trials in PDAC patients reported similar results, as well [51]. Interestingly, margin-negative resection rates were significantly more common following neoadjuvant treatment in these clinical trials. Sadly, mesopancreatic fat infiltration status was not included in the analysis of these patients.

In a larger study addressing PDAC patients after upfront surgery, preoperative MDCT was a viable radiographic tool to predict mesopancreatic fat infiltration and resection margin status [23] strengthening the argument that neoadjuvant therapy leads to increased fibrosis even if the response is not sufficient to eradicate all vital tumor cells. There was a clear trend between radiographically presumed and histopathologically verified tumor response, which did not reach statistical significance due to the small number of enrolled patients. Yet, patients without MDCT presumed tumor response were significantly more prone to mesopancreatic fat infiltration. Thus, pre-operative MDCT staging did predict true tumor extensions in neoadjuvantly treated patients, as well [34].

## 5. Conclusions

Vital tumor cells were present in the retropancreatic fatty tissue following neoadjuvant therapy for PDAC in more than 60% of patients. Thus, mesopancreatic excision (MPE) is still justified even after neoadjuvant treatment. Neoadjuvant treatment and MPE were able to increase margin-negative resection rates when compared to patients receiving upfront MPE. Mesopancreatic fat infiltration was observed to be independent from histopathological tumor response, indicating that mesopancreatic fat infiltration may be a sign of adverse tumor topography, rather than advanced tumor biology. Patients with MDCT-predicted tumor response were less prone to mesopancreatic fat infiltration, underlining the importance of mesopancreatic fat stranding in neoadjuvant treated patients as well. MPE and the evaluation of mesopancreatic fat stranding by MDCT should, therefore, be the standard of care following neoadjuvant treatment regimens for PDAC.

## Figures and Tables

**Figure 1 cancers-14-00068-f001:**
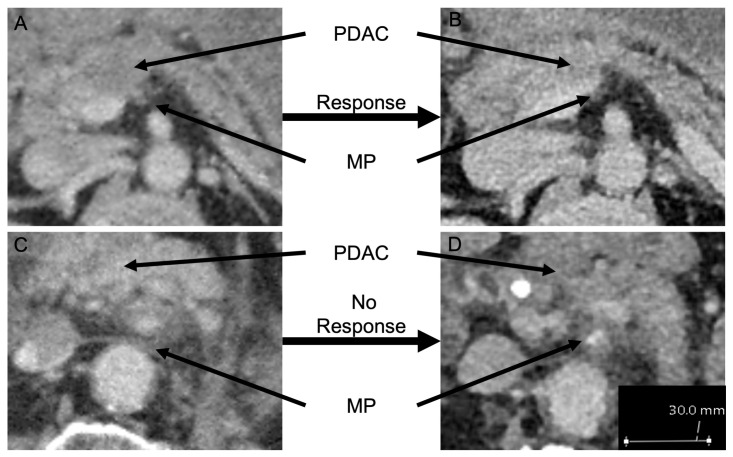
MDCT slides of PDAC patients prior to and after neoadjuvant therapy (Patient 1: slide (**A**,**B**), Patient 2: slide (**C**,**D**)). (**A**) MDCT slide prior to neoadjuvant therapy with MPS. (**B**) MDCT slide after neoadjuvant therapy and treatment response without MPS. (**C**) MDCT slide prior to neoadjuvant therapy with MPS. (**D**) MDCT slide after neoadjuvant therapy without treatment response. (MDCT: multiphasic computed tomography, MPS: mesopancreatic fat stranding). Patient 1: ypT2N1M0G2R0CRM−; Patient 2: ypT3N1M0G3R1.

**Figure 2 cancers-14-00068-f002:**
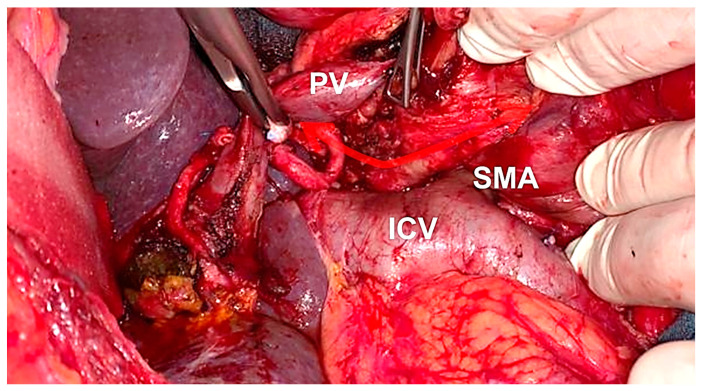
Intraoperative picture of situs during MPE. Note the strict dissection around the caval vein (ICV) and abdominal aorta until the origin of the superior mesenteric artery (SMA) and celiac trunk (red arrow). (ICV, inferior caval vein, PV: portal vein, SMA: superior mesenteric artery).

**Figure 3 cancers-14-00068-f003:**
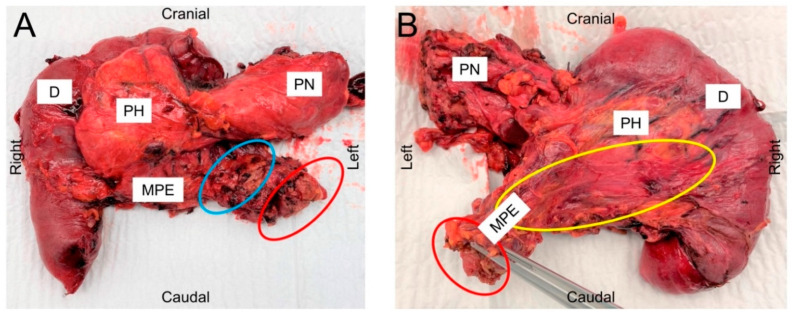
(**A**) Ventral view of specimen following pancreaticoduodenectomy for PDAC demonstrating mesopancreatic excision. (**B**) Posterior view of specimen. Note the fibrous tissue in B (yellow circle) extending between the mesenteric origin of the superior mesenteric artery and the duodenum. Positional markings indicate the position of specimen in situ. Red circle: pedicle of the mesopancreas arising from the SMA; blue circle: medial groove of the portal vein; yellow circle: Treitz fascia dissected and attached to the dorsal resection margin running up to the pedicle of the mesopancreas. (D: duodenum; MPE: mesopancreatic excision PH: pancreatic head; PN: pancreatic neck).

**Figure 4 cancers-14-00068-f004:**
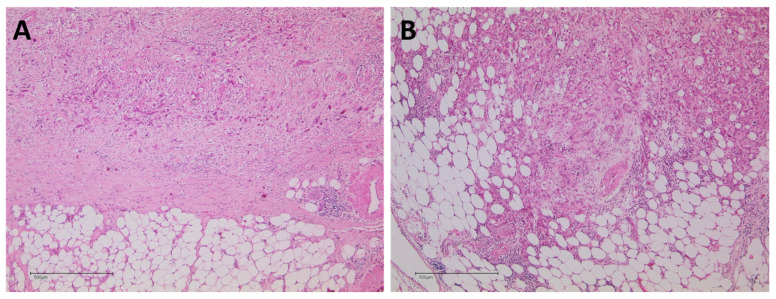
(**A**) Mesopancreatic fatty tissue without PDAC infiltration (H&E, 5×). (**B**) Mesopancreatic fatty tissue with abundant PDAC infiltration (H&E, 5×).

**Figure 5 cancers-14-00068-f005:**
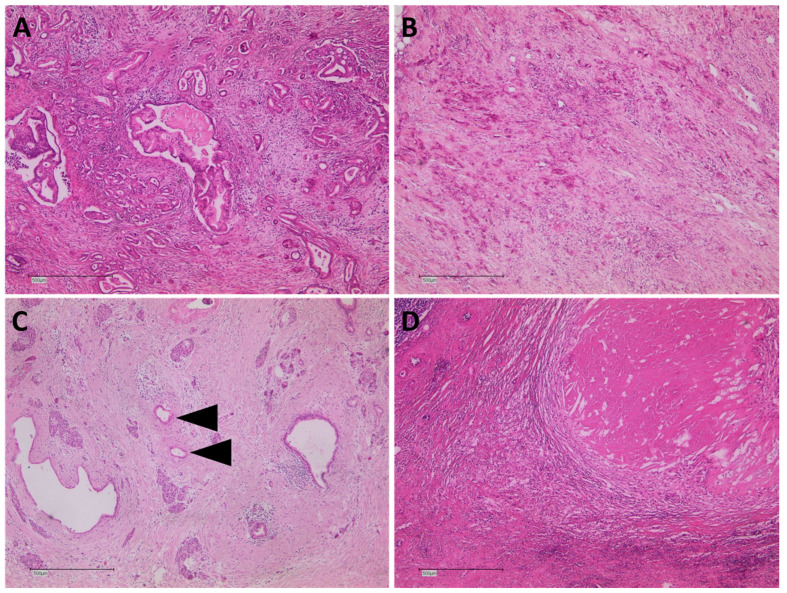
Spectrum of tumor regression grading in pancreatic cancer. (**A**) PDAC with CAP 3 displays abundant vital residual tumor with nearly no regressive changes (H&E, 5×). (**B**) PDAC with CAP 2 shows partial regression with collagen-rich fibrosis and inflammatory infiltrate, but vital residual tumor exceeding rare small groups of tumor cells (H&E, 5×). (**C**) PDAC with CAP 1 is characterized by near-complete response showing only rare single tumor glands embedded in vast collagen-rich fibrosis with residual normal pancreatic tissue (H&E, 5×). (**D**) PDAC with CAP 0 equals complete response with necrosis, fibrosis and inflammatory resorption, but no vital tumor cells (H&E, 5×).

**Table 1 cancers-14-00068-t001:** Demographic table of all 27 patients of the neoadjuvant cohort. Staging is revised to the 8th Edition of the UICC TNM classification of malignant tumors.

Age in Years		
Median (Range)	66 Years (41–80)	
Sex	*n*	%
Male	16	59.3
Female	11	40.7
T-stage		
ypT0	2	7.4
ypT1	3	11.1
ypT2	13	48.1
ypT3	9	33.3
N-stage		
N0	11	40.7
N1	10	37
N2	6	22.2
Grading		
G1/G2	18	66.6
G3	9	33.3
Pn		
Pn0	9	33.3
Pn1	18	66.6
L		
L0	20	74.1
L1	7	25.9
V		
V0	21	77.8
V1	6	22.2
R-status		
R0(CRM−)	17	62.9
R1/R0(CRM+)	10	37.1
MPI		
Positive	17	62.9
negative	10	37.1

CRM: circumferential resection margin; Hep: hepatic; L: lymphatic invasion; MPI: mesopancreatic fat infiltration; Pn: perineural invasion; V: venous invasion.

**Table 2 cancers-14-00068-t002:** Analysis of patients stratified according to positive and negative mesopancreatic infiltration, *n* = 27. There was a heterogenous distribution of clinico-pathological variables. Statistical significance was calculated by chi-squared test. ** indicates a *p*-value ≤ 0.01; * indicates a *p*-value ≤ 0.05.

	No Mesopancreatic Fat Infiltration		Mesopancreatic Fat Infiltration		*p*-Value
	*n* = 10		*n* = 17		
Treatment response	*n*	%	*n*	%	0.003 **
CAP 0	2	20	0	0	
CAP 1	4	40	1	5.9	
CAP 2	4	40	11	64.7	
CAP 3	0	0	5	29.4	
R-status					0.031 *
R0(CRM−)	8	80	9	52.9	
R1/R0(CRM+)	2	20	8	47.1	

CAP: College of American Pathologists; CRM: circumferential resection margin.

**Table 3 cancers-14-00068-t003:** Analysis of patients stratified according resection status, *n* = 27. Patients without mesopancreatic fat infiltration showed a higher rate of R0CRM− resections. Statistical significance was calculated by chi-squared test. ** indicates a *p*-value ≤ 0.01.

	R0(CRM−)		R1/R0(CRM+)		*p*-Value
	*n* = 17		*n* = 10		
Treatment Response	*n*	%	*n*	%	0.042 **
CAP 0	2	11.8	0	0	
CAP 1	5	29.4	0	0	
CAP 2	9	52.9	6	60	
CAP 3	1	5.8	4	40	

CAP: College of American Pathologists; CRM: circumferential resection margin.

**Table 4 cancers-14-00068-t004:** Analysis of patients stratified according to MDCT-predicted tumor response, *n* = 27. MP infiltration status was distributed heterogeneously across MDCT tumor response status. Statistical significance was calculated by chi-squared test. * indicates a *p*-value ≤ 0.05.

	MDCT Tumor Response		MDCT No Tumor Response		*p*-Value
	*n* = 17		*n* = 10		
Treatment response	*n*	%	*n*	%	0.122
CAP 0 and 1	6	35.3	1	10	
CAP 2 and 3	11	64.7	9	90	
MP Infiltration					* 0.042
positive	8	47.1	9	90	
negative	9	52.9	1	10	
R-status					0.692
R0(CRM−)	11	64.7	6	60	
R1/R0(CRM+)	6	35.3	4	40	

CAP: College of American Pathologists; CRM: circumferential resection margin; MDCT: multi-detector computed tomography; MP: mesopancreatic.

**Table 5 cancers-14-00068-t005:** Analysis of patients stratified according to neoadjuvant therapy followed by surgery (*n* = 27) vs. upfront surgery *n* = 173. Mesopancreatic fat infiltration was heterogeneously distributed across the sub-groups. Statistical significance was calculated by chi-squared test.

MP Status	Neoadjuvant and Surgery		Upfront Surgery		*p*-Value
	*n* = 27		*n* = 173		
MP Infiltration					0.039
positive	17	62.9	131	75.7	
negative	10	37.1	42	24.3	

MP: mesopancreatic.

**Table 6 cancers-14-00068-t006:** Analysis of metachronous disease stratified according to treatment constellation. Rate of systemic relapse was similar between neoadjuvant and upfront surgery-treated patient groups (*p* = 0.143; not shown). Local tumor control was significantly improved after neoadjuvant treatment when compared to patients who received upfront surgery (*p* = 0.040). Statistical significance was calculated by chi-squared test.

Therapy Modality	No Metastases		Systemic Relapse		Local Recurrence		*p*-Value
	*n*	%	*n*	%	*n*	%	
Neoadjuvant	5	18.5	18	66.7	2	7.4	0.04
*n* = 27							
Upfront surgery	63	36.4	81	46.8	29	16.8	
*n* = 173							

## Data Availability

The datasets used and/or analyzed during the current study are available from the corresponding author on reasonable request.

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
