# Peer review of "Neoadjuvant Treatment Lowers the Risk of Mesopancreatic Fat Infiltration and Local Recurrence in Patients with Pancreatic Cancer"

_cancers, 2021, doi:10.3390/cancers14010068_

Round 1
Reviewer 1 Report
Radiographic, clinicopathological and survival parameters of 26 consecutive patients who underwent neoadjuvant therapy prior to MPE were evaluated. The paper use of CT in evaluating therapeutic response and mesenteric fat infiltration in pancreatic cancer is relatively innovative.There are the following points to be modified.
- No specific dosing regimen of neoadjuvant chemotherapy was indicated
- The sample size included in the study is small
- The scoring criteria for CT evaluation of treatment response and mesangial infiltration were not indicated
- The author only made a comparison between groups and did not make a correlation analysis. Please modify the description of correlation analysis in the article
- After neoadjuvant therapy, what is the proportion of imaging evaluation from unresectable to resectable
- In resection status part(282-288 lines), only 11 patients (42.3%) were evaluated CRM:(circumferential resection margin).However, 26 patients were evaluated CRM in the full text, please explain in detail
- Table 6 is not clearly explained in lines 342-343, please modify it as appropriate
- Did the study investigate the relationship between fatty infiltration and recurrence?
- The study lacked analysis of prognostic survival data.
Author Response
Radiographic, clinicopathological and survival parameters of 26 consecutive patients who underwent neoadjuvant therapy prior to MPE were evaluated. The paper use of CT in evaluating therapeutic response and mesenteric fat infiltration in pancreatic cancer is relatively innovative. There are the following points to be modified.
- No specific dosing regimen of neoadjuvant chemotherapy was indicated
We thank the reviewer for this insightful comment. We now stated the dosing for each regimen (Supplemental Table i) (Line 267).
- The sample size included in the study is small
We agree with the reviewer that it would be advantageous to have a bigger sample size. During the submission process, we were able to enroll one additional patient. However, since neoadjuvant treatment is not yet standard in the care of pancreatic cancer patients, the number of treated patients is limited. However, compared to other retrospective mono-institutional studies, a similar amount of patients was enrolled in our analysis (1). As this new form of treatment is still under investigation and current clinical trials have not demonstrated a significantly improved survival of neoadjuvantly treated borderline resectable patients when compared to patients after upfront surgery (2, 3). We revised the discussion section to address this concern and included the limited sample number as a limitation of our study.
This reads as followed: One limitation of our study is the small cohort. Yet, when compared to other recent retrospective studies on neoadjuvant treatment (50, 51), a similar number of patients were enrolled as in our analysis. Another limitation of our study is the retrospective and mono-institutional nature of the analysis. Patients received different neoadjuvant regimes in this study, further limiting our conclusions due to heterogeneity. However, as neoadjuvant treatment is still under investigation and current clinical trials have not demonstrated a significantly improved survival of neoadjuvantly treated borderline resectable patients when compared to patients after upfront surgery (32, 46, 52), the number of eligible patients is still limited. Furthermore, it remains unknown which multimodal regime is superior to the other in a neoadjuvant setting. Taking this small sample size into consideration, we did not perform a survival analysis, as its statistical relevance would be questionable. Nevertheless, a negative association was found between neoadjuvant treatment and local recurrence, concluding that multimodal therapy following MPE could contribute to local tumor control.
- The scoring criteria for CT evaluation of treatment response and mesangial infiltration were not indicated
The scoring system has now been described in detail, as previously reported here (6). This reads as followed: Scoring criteria of tumor response was scored by the mesopancreatic attenuation of the fatty tissue in terms of diminished/decreased mesopancreatic fat stranding and/or de-creased tumor size with an smaller encasement of the SMA or celiac trunk/common hepatic artery. Due to the limited number of enrolled patients, no subgroup analyses (MPS1-3) was performed. Patients were subgrouped into tumor response or no tumor response.
- The author only made a comparison between groups and did not make a correlation analysis. Please modify the description of correlation analysis in the article
We thank the reviewer for pointing out this mistake. Indeed, we did not perform correlation analyses in all tables and have now corrected the material and methods section accordingly. When analyzing the different distribution in two groups (Table 2-6) we performed a distribution analysis using the chi squared test. In the case of the mesopancreatic infiltration in regard to the treatment response (Tab 2), we have now also performed a correlation analysis (Pearson`s correlation coefficient), which is now included in the supplemental materials (Suppl Tab ii).
- After neoadjuvant therapy, what is the proportion of imaging evaluation from unresectable to resectable
Of the cohort, 17 patients with MDCT evaluated tumor response were re-staged from unresectable to resectable. The other 9 patients were still deemed non-resectable upon radiographic staging following neoadjuvant therapy, yet did present with a stable disease. Surgical exploration to test for secondary resectability was performed in these cases, as is proposed now in our pending national guidelines. While not reaching statistical significance, we observed a lower rate of R0CRM- resections in these patients without MDCT tumor response, compared to the patients with MDCT tumor response (Table 4). Since resection status is dependent on multifactorial biases, more importantly a positive mesopancreatic fat infiltration status was less prone in patients with MDCT predicted tumor response. The discussion section was revised as well.
- In resection status part(282-288 lines), only 11 patients (42.3%) were evaluated CRM:(circumferential resection margin).However, 26 patients were evaluated CRM in the full text, please explain in detail
We thank the reviewer for pointing out this ambiguity in our explanation. As stated in the material and method section and the results, patients who received surgery before 2015 were not initially assessed according to the CRM-status and required a histopathological re-analysis to evaluate CRM. Histopathological slides were re-visited to reproduce the CRM status, resulting in the cohort of 27 patients with complete CRM status. We have tried to state this more clearly in the manuscript now.
- Table 6 is not clearly explained in lines 342-343, please modify it as appropriate
Table 6 was now explained in detail and results from table 5 were incorporated for a conclusive statement. The discussion section was revised accordingly.
- Did the study investigate the relationship between fatty infiltration and recurrence?
No, we did not investigate a direct correlation between fatty infiltration and local or systemic recurrence. We did, however, observe that neoadjuvant treatment decreased the rate of mesopancreatic fat infiltration, as well as the rate of local recurrences. One might thus assume, taking into consideration the biology of pancreatic cancer, that a higher rate of fat infiltration may facilitate higher rates of local recurrences.
- The study lacked analysis of prognostic survival data.
The aim of this study was not to investigate the survival benefit of neoadjuvant treatment. Recent larger, randomized trials investigating this have failed to demonstrate a significant survival improvement in patients following neoadjuvant treatment compared to patients after upfront surgery. We here focus on clinicopathological factors that may influence surgical treatment of patients receiving neoadjuvant treatment.
Literature:
- Hu Q, Wang D, Chen Y, Li X, Cao P, Cao D. Network meta-analysis comparing neoadjuvant chemoradiation, neoadjuvant chemotherapy and upfront surgery in patients with resectable, borderline resectable, and locally advanced pancreatic ductal adenocarcinoma. Radiation oncology (London, England). 2019;14(1):120.
- Versteijne E, Suker M, Groothuis K, Akkermans-Vogelaar JM, Besselink MG, Bonsing BA, et al. Preoperative Chemoradiotherapy Versus Immediate Surgery for Resectable and Borderline Resectable Pancreatic Cancer: Results of the Dutch Randomized Phase III PREOPANC Trial. Journal of clinical oncology : official journal of the American Society of Clinical Oncology. 2020;38(16):1763-73.
- Janssen QP, van Dam JL, Bonsing BA, Bos H, Bosscha KP, Coene P, et al. Total neoadjuvant FOLFIRINOX versus neoadjuvant gemcitabine-based chemoradiotherapy and adjuvant gemcitabine for resectable and borderline resectable pancreatic cancer (PREOPANC-2 trial): study protocol for a nationwide multicenter randomized controlled trial. BMC cancer. 2021;21(1):300.
- Rutter CE, Park HS, Corso CD, Lester-Coll NH, Mancini BR, Yeboa DN, et al. Addition of radiotherapy to adjuvant chemotherapy is associated with improved overall survival in resected pancreatic adenocarcinoma: An analysis of the National Cancer Data Base. Cancer. 2015;121(23):4141-9.
- Motoi F, Kosuge T, Ueno H, Yamaue H, Satoi S, Sho M, et al. Randomized phase II/III trial of neoadjuvant chemotherapy with gemcitabine and S-1 versus upfront surgery for resectable pancreatic cancer (Prep-02/JSAP05). Japanese journal of clinical oncology. 2019;49(2):190-4.
- Safi SA, Haeberle L, Heuveldop S, Kroepil P, Fung S, Rehders A, et al. Pre-Operative MDCT Staging Predicts Mesopancreatic Fat Infiltration-A Novel Marker for Neoadjuvant Treatment? Cancers. 2021;13(17).
Reviewer 2 Report
The present study aimed at analyzing the warranty of mesopancreatic excision in pancreatic cancer after neoadjuvant treatment. The study shows that after neoadjuvant treatment, both the rate of R0 resection R0(CRM-) and the rate of mesopancreatic fat infiltration was 61.5%, indicating that neoadjuvant therapy alone may not be sufficient for local tumor control. The authors concluded that MPE is still warranted after neoadjuvant treatment.
Despite an interesting and original hypothesis, many limitations jeopardize the global message of the paper
Comments
Scientific and methodological accuracy of the paper
- Despite an interesting and original hypothesis, this study failed to demonstrate any benefit because of limitation of sample size and heterogeneity of selected populations. Indeed, concerns about margins in PDAC located in the head are totally different than those located in the body / tail for anatomical reasons. For example, the authors should mention whether the Gerota fascia was routinely removed in MPE and how extended was the dissection. They should select only patients undergoing pancreatoduodenectomy with a standardized lymphadenectomy and a standardized evaluation of surgical margins.
- Additionally, Impact of neoadjuvant treatment of peritumoral fat is clearly different when gemcitabine monotherapy, combined therapy with gemcitabine or radio-chemotherapy including gemcitabine are used. Specifically, radio-chemotherapy routinely induces infiltration of fat within the treated area. Could the authors please explain how they interpreted radiotherapy-induced infiltration?
- Another significant point is the complex association between positive margins specifically on the retroperitoneal layer and occurrence of local recurrence. Unfortunately, the results of the present series cannot draw any conclusion about modification of surgical strategy according to tumor response on imaging.
Missing data that would make the paper complete and more comprehensive
- A figure of CT scan illustrating the modification of peritumoral fat before of after neoadjuvant treatment would improve the clarity of the manuscript. A correlation between imaging and pathology tumor response would be interesting.
- The authors should significantly develop the limitation section of their study. Please take in account comments mentioned above.
- Some interesting references could be added:
Rutter CE, Park HS, Corso CD, Lester-Coll NH, Mancini BR, Yeboa DN, Johung KL. Addition of radiotherapy to adjuvant chemotherapy is associated with improved overall survival in resected pancreatic adenocarcinoma: An analysis of the National Cancer Data Base. Cancer. 2015 Dec 1;121(23):4141-9. doi: 10.1002/cncr.29652. Epub 2015 Aug 17. PMID: 26280559.
Hu Q, Wang D, Chen Y, Li X, Cao P, Cao D. Network meta-analysis comparing neoadjuvant chemoradiation, neoadjuvant chemotherapy and upfront surgery in patients with resectable, borderline resectable, and locally advanced pancreatic ductal adenocarcinoma. Radiat Oncol. 2019 Jul 10;14(1):120. doi: 10.1186/s13014-019-1330-0. PMID: 31291998; PMCID: PMC6617703.
Author Response
The present study aimed at analyzing the warranty of mesopancreatic excision in pancreatic cancer after neoadjuvant treatment. The study shows that after neoadjuvant treatment, both the rate of R0 resection R0(CRM-) and the rate of mesopancreatic fat infiltration was 61.5%, indicating that neoadjuvant therapy alone may not be sufficient for local tumor control. The authors concluded that MPE is still warranted after neoadjuvant treatment.
Despite an interesting and original hypothesis, many limitations jeopardize the global message of the paper
Comments
Scientific and methodological accuracy of the paper
- Despite an interesting and original hypothesis, this study failed to demonstrate any benefit because of limitation of sample size and heterogeneity of selected populations. Indeed, concerns about margins in PDAC located in the head are totally different than those located in the body / tail for anatomical reasons. For example, the authors should mention whether the Gerota fascia was routinely removed in MPE and how extended was the dissection. They should select only patients undergoing pancreatoduodenectomy with a standardized lymphadenectomy and a standardized evaluation of surgical margins.
We agree with the reviewers concern relating to the standardized surgical resection of pancreatic cancer. We have now included more information regarding our standard approach in PDAC of the pancreas head. All patients included in this analysis underwent surgery according to our resection standards, including lymphadenectomy, as stated in the material and method section. As now stated in the manuscript, during mesopancreatic excision, Gerota´s fascia is transected from laterally. Most importantly Toldt´s fascia from the ascending colon is also transected until the dorsal border of the duodenum and this transection line is continued with the dissection of Treitz´ fascia medially behind the duodenum until the Treitz ligament and the origin of the SMA.
- Additionally, Impact of neoadjuvant treatment of peritumoral fat is clearly different when gemcitabine monotherapy, combined therapy with gemcitabine or radio-chemotherapy including gemcitabine are used. Specifically, radio-chemotherapy routinely induces infiltration of fat within the treated area. Could the authors please explain how they interpreted radiotherapy-induced infiltration
We thank the reviewer for this insightful comment. In the current literature, the mesopancreatic fat status in PDAC patients was only marginally studied. No literature is available on the histopathologic mesopancreatic fat infiltration status following neoadjuvant treatment. The radiologically observed stranding of the mesopancreatic fatty tissue following radiotherapy is indeed very interesting, yet a distinction between viable tumor cells and unspecific radio-inflammation is not possible radiographically. Our results are thus even more interesting, as we are the first to explore and correlate these clinical findings. Furthermore, in our hospital, radiotherapy for PDAC patients is not routinely used and only 3 of the patients included here received neoadjuvant radiochemotherapy.
- Another significant point is the complex association between positive margins specifically on the retroperitoneal layer and occurrence of local recurrence. Unfortunately, the results of the present series cannot draw any conclusion about modification of surgical strategy according to tumor response on imaging.
We thank the reviewer for this very important point. We agree that we cannot prove a direct link between local recurrence and neoadjuvant treatment. Here, we merely found that neoadjuvant treatment was associated with a significantly lower MP infiltration status and lower rate of local recurrence. Margin negative resection rates were increased when compared to the upfront surgery patient group, but did not reach statistical significance. As correlation is no causation, we cannot prove that there is indeed causation in this observation. But the biology of the disease and clinical experience make this assumption more than a leap of faith. We have amended our manuscript to include the reviewers concerns.
This reads as followed:
One limitation of our study is the small cohort. Yet, when compared to other recent retrospective studies on neoadjuvant treatment (50, 51), a similar number of patients were enrolled as in our analysis. Another limitation of our study is the retrospective and mono-institutional nature of the analysis. Patients received different neoadjuvant regimes in this study, further limiting our conclusions due to heterogeneity. However, as neoadjuvant treatment is still under investigation and current clinical trials have not demonstrated a significantly improved survival of neoadjuvantly treated borderline resectable patients when compared to patients after upfront surgery (32, 46, 52), the number of eligible patients is still limited. Furthermore, it remains unknown which multimodal regime is superior to the other in a neoadjuvant setting. Taking this small sample size into consideration, we did not perform a survival analysis, as its statistical relevance would be questionable. Nevertheless, a negative association was found between neoadjuvant treatment and local recurrence, concluding that multimodal therapy following MPE could contribute to local tumor control.
Our results emulate similar observations as in studies with rectal cancer patients. The introduction of neoadjuvant radiochemotherapy and the standardization of total mesorectal excision (19), performed independent of the tumor response, resulted in significantly lower local recurrence rates, while overall survival remained similar to patients undergoing upfront surgery (41, 53, 54). Recently published randomized trials in PDAC patients reported similar results, as well (52). Interestingly, margin negative resection rates were significantly more common following neoadjuvant treatment in these clinical trials. Sadly, mesopancreatic fat infiltration status was not included in the analysis of these patients.
Missing data that would make the paper complete and more comprehensive
- A figure of CT scan illustrating the modification of peritumoral fat before and after neoadjuvant treatment would improve the clarity of the manuscript. A correlation between imaging and pathology tumor response would be interesting.
We thank the reviewer for raising this important point, figures were now added (Figures 1A-D).
- The authors should significantly develop the limitation section of their study. Please take in account comments mentioned above.
The discussion section was revised.
The limited amount of enrolled patients was addressed as one of the major concerns.
- Some interesting references could be added:
We thank the reviewer of these relevant and interesting additional references. These references were taken now added.
Round 2
Reviewer 2 Report
This revised version is significantly improved
The authors should be congratulated